# Indexing Deficiency: Connecting Language Learning and Teaching to Evaluations of US Spanish

Gabriella Licata 



Latino and Latin American Research and Studies Center, University of California, Riverside, CA 92521, USA; glicata@ucr.edu

**Abstract:** The examination of language attitudes towards US Spanish variables unearths indexical meanings rooted in deficit perspectives, particularly in educational contexts. Standard language ideologies undergird pedagogical practice and learning experiences in second language (L2) and heritage language (HL) Spanish classes. The present study utilizes dual research paradigms of social cognition (matched guise technique (MGT); implicit association test (IAT)) to determine if varying experiences with (Spanish) standard language ideologies in academic settings condition bias towards standardized Spanish (SS) and US Spanish (USS) repertoires. L2 and HL students as well as teachers of Spanish ($n = 81$) have more positive associations of SS in both the MGT and IAT, demonstrating that standard language ideologies influence perceptions of language acquisition and academic language learning. No correlations between the bias measures were reported yet attitudes did not differ, suggesting that attitudes are stable and reflected in both early learnings of social information and lived experiences throughout formative education. These results contribute to a growing body of research that examines how monoglossic ideologies reinforce and reproduce the stigma associated with features of US Spanish(es).

**Keywords:** heritage languagers; deficit perspectives; raciolinguistics; social cognition; language education



## 1. Introduction

The so-called inherent relationship between standardized language (i.e., an idealized, hypothetical construct of accepted language use; see (Crowley 2003); Lippi-Green 2012) and academic registers has undergirded how language education in the United States is organized (Flores 2020a; Rosa and Flores 2017), despite the fact that a universal acceptance of what 'academic language' is does not exist (R. A. Martínez and Mejía 2020). With this in mind, the indexical field (see Eckert 2008; Silverstein 2018) of US Spanish (commonly referred to as Spanglish; see Otheguy and Stern 2011) in educational contexts is conditioned by what is institutionally considered to be "appropriate" or "academic" (Rosa and Flores 2017). These registers may well be defined by what they are *not*; that is, teachers and curriculum designers may decide what language is suitable in a given context, and thus certain sounds, lexical items, and phrasing considered 'inappropriate' may be viewed unfavorably in academic contexts where they are deemed unsuitable by educators and students alike. A rejection of what Bakhtin (1981) termed *heteroglossia*—the simultaneous use of different kinds of forms or signs (Bailey 2012)—privileges monoglossia and legitimizes the monolingual languager[1] as well as linguistic variables indexing standardized forms (García 2011, p. 115). The standardizations of language production and learning are presented as monoglossic objects that can be impartially measured through standardized testing (Siordia and Kim 2022); however, the over-representation of racialized students in language remediation programs demonstrates how implicit biases condition the perception of their home varieties, as the assessment itself is "reflective of larger settler colonial frameworks embedded in linguistic standards that continue to drive education and language ideologies/practices globally. . . especially in U.S. schools" (Cioè-Peña 2022, p. 25).

These ideological assumptions generally condition the formative experiences of multilingual youth in US schools whose first or home language is deemed less prestigious or less useful by hegemonic standards. The institutional negation of their dynamic language abilities results in a process that co-constructs the pathologization of their communication and the racialization of their identities (Rosa and Flores 2017). These damaging co-constructions of a student and their linguistic repertoire may be mediated by other social factors, including gender and economic class (Solorzano and Yosso 2001). Resultantly, raciolinguistic ideologies are hegemonically sustained, as white-perceiving subjects (Rosa and Flores 2017) view the indexical meanings of Latinx students' language repertoires through deficit lenses.

The results of these ideologies manifest in the homogenization of language teaching and learning, paving the way for the commodification of language through a standardized scope and sequence of grammar and vocabulary (VanPatten 2015, p. 10) that conceptualizes language variation as exceptional and is marked in comparison to the standardized repertoire, which is more fixed (Train 2020). As such, Spanish as a language of instruction in the United States has undergone a hyperstandardization (Train 2003) that ignores empirically observed community use of language, particularly varieties in the United States. For example, in Northern California (and beyond), the term *rentar* and *alquilar* ("to rent") are synonymous, but the latter is the term often found in textbooks as the standardized or academic option with perhaps *rentar* denoted as a 'colloquial' or 'informal' option, though the latter is in widespread use in many domains. As a result, students who grow up using Spanish in the household are expected to 'master' an academic or prestige variety that stands in contrast to their home and community language practices, which in many regards depicts the latter varieties as 'incomplete' or 'insufficient' (see Rosa and Flores 2017).

The separation of home from school varieties upholds standard language ideologies and also relies on the notion that students can and should keep these repertoires separate. The distinctiveness of named languages—varieties that undergo codification and are labeled (often related to nation building; see García 2019)—is used to create language boundaries in the classroom, which ignores a "usage-based account of language instead which does not take linguistic systems for granted but construes them based on the single linguistic unit" (Wasserscheidt 2021). This damaging discourse has undergirded research concerning US Spanish languagers while ignoring the dynamic and lived reality of *translanguaging*, or when one utilizes their full linguistic repertoire in communication (García and Otheguy 2020; García et al. 2021). While codeswitching examines how multiple named languages may be used in an utterance, translanguaging is "a perspective that prioritizes the meaning-making of bi/multilinguals in ways that challenge normative language ideologies" (Flores 2020b).

As a result, multilingual students' abilities to dynamically language are ignored and undervalued as demonstrative of communicative ability because they do not communicate within the bounds of named languages. Thus, heritage language[2] programs are often geared towards preparing students to achieve double monolingualism (Heller 2006; Rosa and Flores 2017). Second language programs alike aim to have students develop a hyperstandardized "language-elsewhere"(Mena 2022), or rather a variety of Spanish disembodied from the surrounding communities. In understanding how these damaging ideologies are epistemologically situated in the classroom and in language learning, experimental sociolinguistics rooted in anthropological frameworks that investigate language ideologies can inform us of the explicit and implicit biases present in education. In viewing schools as one of the principal sites where damaging language ideologies are born and perpetuated, I investigate how early learned bias and Spanish language training may influence perceptions of US Spanish features that are historically stigmatized to index 'deficits.' The present study utilizes dual models of social cognition to determine if the differentiated experiences of second language learners, heritage learners, and Spanish language teachers have differing bias towards standardized Spanish and US Spanish repertoires. This experimental line combines the matched guise technique (Lambert et al. 1966; MGT) and implicit association test (Greenwald et al. 1998; IAT) to determine if any groups are

exhibiting attitude shifts via a divergence of more malleable attitudes (elicited in the MGT) from deeply rooted implicit bias (elicited in the IAT). Findings reveal that all groups' bias measures favor standardized Spanish over US Spanish, assigning, to the former, positive qualities and associations related to complete acquisition and academic quality. However, bias measures did not correlate, despite their similar trajectories, which indicates that in this study, explicit attitudes elicited from the MGT are stable and similar to those elicited from the IAT, despite being processed differently.

Results offer new insights towards the deficit perspectives that plague US Spanish languagers and the lack of legitimization of US Spanish(es) in academic settings. Accordingly, I urge educators to re-examine how traditional models of language teaching maintain raciolinguistic ideologies and deficit perspectives in and outside of the classroom.

## 2. Colonial Underpinnings of Language Separateness

A brief examination of colonial language epistemologies aids us in understanding how multilingual languagers are framed through deficit perspectives. Initial linguistic research legitimized US Spanish languagers and their repertoires by developing a typology of codeswitching (Poplack 1980). This research was groundbreaking at the time, as it affirmed that US Spanish languagers (and other multilinguals) could crosslinguistically navigate Spanish and English with ease, demonstrating that grammatical constraints of more than one language could be learned and employed simultaneously. However, the countless sociolinguistic and anthropological studies that followed and sought to strengthen positive positions of multilingualism have not thwarted the deficit perspectives that plague racialized students in the US (Rosa and Flores 2017).

These views rely on an idealization of codified language, or standard language ideology, defined as "bias toward an abstracted, idealized, homogeneous spoken (and written) language which is imposed and maintained by dominant bloc institutions" (Lippi-Green 2012, p. 64), which in the US is undergirded by the politicization and economization of language. A prominent tool of unabashed colonialism, standard language ideology is a "social construct of the nation-border" (Martínez 2003), reinforcing language boundaries and rejecting the naturalness of translanguaging through the intentional subordination of nonhegemonic and/or nonstandardized languages. Resultantly, these constructs erase linguistic and cultural heterogeneity, facilitating the convenient placement of languages and people into specific demographic categories, and serve as remnants of colonial organization (also known as coloniality of power; see Quijano 2000) that sought and continue to maintain the compartmentalization and homogenization of languages to elevate the idealized speaker.[3] This invalidates those whose forms of communication are weaponized, mocked, and potentially erased, resulting in monoglossic ideologies that are crystallized and interwoven into hegemonic institutions (Rosa and Flores 2017).

The limitations to understanding multilingual languagers' dynamic process result in models of separateness that attempt to codify a languager's production by how frequently they use one variety or the other, such as the matrix language frame (see Myers-Scotton 1997). Socio- and psycholinguistic researchers who describe the alleged 'deficits' of bi- or multilinguals position monolingualism as the norm and Spanish and English as distinct structural and cognitive entities, a process that reproduces and reiterates the identification of people and places with bordered territories (García 2019; Otheguy et al. 2015). The epistemological positioning of languages as separate cognitive systems paves the way for the theorization of multilinguals as *semilingual*, meaning that they have not fully acquired one or more of the named languages under examination (Martin-Jones and Romaine 1986), and many language programs seek to 'remedy' this supposed deficit.

*Spanish Language Education in the United States*

A deep examination of Spanish language learning models in US contexts reveals colonial epistemologies that situated Spanish language teaching and learning programs in ideologies of appropriateness in both heritage and L2 classes. Burns (2018) examines Span-

ish 'foreign' language pedagogy (textbooks and direct instruction) in a research university, finding that the reproduction of standard language ideologies and variation erasure served to, at times, explicitly delegitimize US Spanish. The very label of 'foreign' ignores and erases the long history of the Spanish language in present-day US territories (Lozano 2018; Train 2007). Similarly, Spanish language education is designed and modeled around an idealized white speaker (Flores and Rosa 2022), as language at the intersection with *latinidad* often exceptionalizes Blackness and indigeneity and erases colonial histories (Chávez-Moreno 2021), which confounds representation in course materials (Anya 2020, 2021; Austin 2022; de los Heros 2009; Padilla and Vana 2022). The consequences of these epistemologies have resulted in the homogenization of Spanish language teaching that privileges the "language-elsewhere" (Mena 2022), and US Spanish languagers who deviate from this academic hyperstandard (Train 2007) are often unjustly classified as having less social and economic capital (Alonso and Villa 2020), suffering from "word gaps" (Avineri et al. 2015), being designated 'long-term English learners' (Flores et al. 2015), and not 'belonging' to any culture or place (Anzaldúa 1987), reiterating white supremacy on every front.

As such, the damage of monoglossia and deficit perspectives affects those who are racialized by hegemonic systems that feign impartiality, for example, standardized testing (Siordia and Kim 2022). Teachers also reiterate and reproduce language ideologies, much in line with the demands of various institutions (e.g., school administrations and federal and state requirements), which trickle down to students and are reiterated in new contexts. However, scholars, language activists, and educators have emphasized the need to move away from these harmful ideologies by dissolving the artificial boundaries between languages and between identities to instead value the natural occurrence of translanguaging and multiculturalism (García et al. 2021). Researchers report on the benefits for students of counter-hegemonic language ideology in heritage language planning (Loza 2017), sociolinguistic awareness and anti-racist training in teacher preparation (Seltzer 2022), and translanguaging in both cognitive and affective domains (Carstens 2016) to reduce deficit perspectives imposed on US Latinx students. Nonetheless, many others maintain that despite decades of teacher training and education, deficit perspectives continue to dominate language learning narratives, maintaining systemic barriers to language access and expression and placing the onus on individuals to change (Flores et al. 2018). The examination of explicit to implicit biases can shed light on these inequities.

## 3. Evaluating Implicit Social Cognition and Linguistic Bias

Quantitative perception studies can elucidate covert language biases that might not otherwise be expressed when participants are explicitly questioned. Studies in sociolinguistic perception have demonstrated that perceiving subjects do not judge speech utterances in isolation, but rather, they develop ideas about language in conjunction with ideas about who is using the language (Rosa and Flores 2017) and the perceived social information associated with the languagers that precedes language judgment (Kang and Rubin 2009). Thus, individual perceiving subjects may reproduce and reiterate broad social stereotypes in their evaluations with little information on the person (see Inoue 2003). Experimental sociolinguistics has sought to reveal biases ranging from those that are explicitly expressed to those that are more automatic or non-verbalizable. Implicit attitudes are considered to be more stable and unchanging, as they are acquired and learned slowly in earlier socialization, while explicit attitudes are deemed more malleable, susceptible to external influence and fast learning, thus regulated, purposeful, and effortful (Petty et al. 2009; Evans 2008; Karpen et al. 2012). To elicit these biases as they relate to language, a variety of attitude assessments have been utilized in sociolinguistic study. Direct methods involve asking participants to answer questions from which explicit bias is displayed. A common and important measure for gauging attitudes, the direct approach is often employed in questionnaire or interview form and is a long-held research paradigm (Garrett 2001, p. 159), and information gathered from direct questioning has been useful in language planning (p. 159).

A common indirect methodology used for gauging attitudes in linguistic research is the *matched guise technique* (MGT; Lambert et al. 1966; Kircher 2016). Participants listen to voices and evaluate them based on a series of social scales (for example, *This person sounds friendly*). The MGT is considered to be an indirect approach to collecting attitudes because participants may be aware that their attitudes are being tested but are not sure of the precise attitudinal object. Thus, they may rate the variants of a linguistic variable differently on a number of characteristics that reveal that one is stigmatized (Solís Obiols 2002). They are also unaware that the attributes they apply to each speaker will later be interpreted as ideological stances towards the linguistic variable (Garrett 2001). MGT studies in small language communities have shown that listeners will issue discrepancies in judgment even when they know that they are listening to the same speaker using different language forms (Soukup 2013) and may even issue more severe ratings when they are made conscious of the fact that they are rating speech with stigmatized language variants (Rosseel et al. 2019).

Attitudinal research assessing both explicit and implicit biases towards US Spanish has shed light on how biases shift depending on the indexical field (see Section 4). In keeping with the notion that "linguistic and social information comes packaged in a single complex signal" (Craft et al. 2020, p. 390), a linguistic variable's indexical field is also shaped by nonlinguistic information perceived during the process of meaning making (Eckert 2008). In fact, perceiving subjects may already make their linguistic judgments of languagers before experiencing their language production, demonstrating how *reverse linguistic stereotyping* (Kang and Rubin 2009) and raciolinguistic ideologies (Rosa and Flores 2017) motivate perceptual categorization. As such, linguists have sought to reveal automatic biases that are less susceptible to manipulation, incorporating the *implicit association test* (IAT; Greenwald et al. 1998; Lane et al. 2007), a research paradigm that estimates the strength of quickly accessed mental associations between concepts (e.g., Spanish/English) and attributes (e.g., Good/Bad) by measuring the differences in response latency to each concept with one of the specific attributes; that is, if a person responds more quickly to the association of English + Good than Spanish + Good, then they have a stronger positive association between the first two.

Different processing models of social cognition help us to understand if there exists a correlation of implicit to explicit attitudes (Lane et al. 2007). As the likelihood of a correlation between different measures of social cognition varies greatly from one environment to another (Fazio and Olson 2003), sociolinguistic research has accordingly exploited this variability of outcomes, demonstrating how spaciotemporal factors may determine a convergence (McKenzie and Gilmore 2017) or a divergence (M. Babel 2010; Pantos and Perkins 2013; Calamai and Ardolino 2020) of evaluations. Those that demonstrate divergence shed light on possible attitude changes in progress (McKenzie and Carrie 2018), exhibiting how linguistic variables can index different meanings depending on the type of social cognitive process accessed. This lends support to the implicit–explicit attitudinal discrepancy hypothesis (IED), which indicates that a divergence of explicit from implicit attitudes can be attributed to "an attitude change in progress at a given point in time" (Karpen et al. 2012; McKenzie and Carrie 2018). The IED hypothesis accounts for the evidence that long-held implicit evaluations generally remain stable, even if explicit attitudes about a concept have changed significantly (e.g., a smoker who no longer smokes may continue to hold positive implicit bias towards the habit). Thus, more divergence of explicit from implicit attitudes indicates that the explicit attitudes under examination are changing from early learned biases, while lower levels of divergence indicate stability across explicit to implicit attitudes (McKenzie and McNeill 2022, p. 21).

Language bias studies assessing explicit and implicit biases have utilized direct and indirect attitude elicitation (respectively) to assess correlation and potential divergences; that is, direct questioning and—most commonly—the IAT, finding weak relations between explicit and implicit attitude results (McKenzie and Carrie 2018). This study aims to understand how attitudes indirectly derived from the MGT diverge from or converge with bias elicited from the IAT to determine stable or changing attitudes among heritage

and L2 learners and teachers. I seek to demonstrate if language learning experiences condition implicit bias towards a standardized Spanish and a US Spanish repertoire, and whether implicit biases are stored differently, providing new insights into the malleability of bias derived from indirect methods and the possibility of experience to condition later learned biases. The next section uses deficit and raciolinguistic perspectives to examine the indexical field of US Spanish features.

## 4. Indexing Deficits

The perceiving subject (Inoue 2003, 2006) does not perceive all Spanish languaging subjects equally. That is, features of US Spanish index different meanings depending on relevant nonlinguistic information that perceivers use to make their judgments, and who is considered to have a linguistic deficit is subject to scrutiny. Perceiving subjects are capable of quickly identifying socially salient linguistic variables via analyses of verbal communication (i.e., signed or spoken language) in conjunction with nonlinguistic social information about the languager (Campbell-Kibler 2009, 2010; Drager and Kirtley 2016; Hay and Drager 2010). The ideologies that bar access to positive associations to an indexical field may be mediated by social factors; that is, societal constructs associated with the languager like gender, economic class, and ethnicity can delimit access to an indexical field, and speakers may (sub)consciously choose to eschew variants in certain instances where they may face scrutiny (see Chappell 2016). For example, if US Spanish languagers are constantly corrected or criticized for their use of so-called broken Spanish, they may avoid the use of US Spanish features in environments where they do not feel safe using them (e.g., the classroom). Thus, the hegemonic pressures that underpin language use can privilege or oppress the linguistic behavior of certain groups, providing some with liberty to navigate the positive meanings of an indexical field and others not.

Deficit perspectives and monoglossic ideologies condition individual and collective attitudes about US Spanish throughout the United States. Rangel et al. (2015) used the matched guise technique to gauge the attitudes of multilingual languagers towards standardized Spanish, standardized American English, and US Spanish in southern Texas, finding that participants rated US Spanish unfavorably. Both standardized American English and standardized Mexican Spanish—when juxtaposed with a codeswitching repertoire—elicited more positive evaluations from participants in the towns of Laredo and Edinburgh (Texas) with regard to solidarity, status, and personal appeal, though participants offered more positive solidarity ratings towards standardized Mexican Spanish over English, demonstrating long-held community ties with a prestige variety of Spanish. The examination of particular linguistic variables within US Spanish(es) also sheds light on the indexical meanings available to languagers. Heritage languagers have demonstrated similar attitudes to monolingual speech with respect to specific variables, such as aspiration (Chappell 2021a). However, in Chappell's (2021b) examination of Mexican American and Mexican languagers' reactions to voices of the aforementioned groups, language experience and identity conditioned perceptions. That is, Mexican Americans had more gradient and nuanced opinions towards concepts like 'bilingualism' and 'language proficiency' than Mexicans, likely due to their experiences navigating different cultural and linguistic communities, while the latter groups' positions were rooted in epistemologies of language separateness and hierarchy that created more exclusive language groups.

Relatedly, identity has been shown to condition perceptions in the classroom. In a matched guise test, both heritage and L2 students rated monolingual Spanish speakers more positively than heritage and L2 speakers, with heritage speakers being described as using the "'least formal' and 'incorrect' variety in comparison to the L2 variety due to dominant stereotypes and ideologies, and the incorporation of lexical characteristics of US Spanish" (Vana 2020, p. ii). This demonstrates how the stigmatization of lexical features of US Spanish, when juxtaposed with standardized features, may be explicitly or implicitly taught to L2 learners. Similarly, as L2 (Spanish) instruction increases attention towards sociolinguistic competence (Van Compernolle and Williams 2012; Sun 2014), understanding

L2 learners' attitudes towards linguistic variables sheds light on how exposure to variation and both linguistic and cultural proficiency can mediate the social salience of variants and attitudes towards them (Chappell and Kanwit 2022). Quan (2020) found that despite the positive effects of integrating Critical Language Awareness (CLA) materials in a Spanish language course for L2 learners (e.g., compassion for Latinx communities), students still explicitly expressed "ongoing agreement with standard language ideologies related to bilingualism and foreign accents" (p. 915), evidencing the importance of CLA in all language courses and a restructuring of language education. Attitudinal study can also be used to critically investigate teacher bias towards the language variation their students employ. In a matched guise test examining teacher attitudes towards syntactic and lexical features of US Spanish, teachers in general expressed negative attitudes towards the utterances, describing them as 'incorrect' even when they were not sure how to 'correct' them (Román et al. 2019). Bilingual teacher candidates in South Texas similarly reproduce harmful standard language ideologies in their own teaching practices. These were revealed to be reiterations that are learned in their home from their mothers (i.e., their language socializers) due to the societal and institutional linguistic violence that they themselves faced in youth (Ek et al. 2013).

As increased attention is given to the incorporation of implicit attitude measures in the study of (US) Spanish(es), linguists have begun incorporating the implicit association test in attitude studies. Callesano and Carter (2022) found that, when Spanish and English were contrasted in the IAT, Miami-based participants more quickly associated English + Good over Spanish + Good, an effect that was mediated by increased time spent in Miami, demonstrating how binaries shift when hegemonies shift. Similarly, Ianos et al. (2020) issued an IAT and direct questionnaire to adolescents to evaluate bias towards Catalan and Spanish. IAT results showed an overall positive bias towards Catalan over Spanish, though positive bias shifted when tested against the adolescent's home language (i.e., Spanish used at home produced positive associations with Spanish). These findings demonstrate how the strength of solidarity and institutional support can combat hegemonic pressures to assimilate at the local or regional level. To date, the IAT research paradigm has not been used to examine biases towards lexical features of US Spanish (or Spanglish), specifically in comparison with explicit attitudes. This study adds to the existing literature by using an innovative research design to evaluate the effects of language training on attitudes towards a standardized Spanish and a US Spanish repertoire, measuring and comparing bias via different models of social cognition. I seek to answer the following questions:

1. Are US Spanish lexical features salient to groups that have different experiences with Spanish language learning, and what do they index in terms of perceived acquisition and academic-ness when juxtaposed with standardized Spanish lexical features?
2. To what extent is differential participation in language education programs (L2 students, heritage students, and teachers) correlated with more positive explicit attitudes toward US Spanish features?
3. Do the attitudes and associations elicited from indirect methods (i.e., the MGT) and automatic response (i.e., the IAT) indicate that explicit and implicit biases result from distinct cognitive processes?

To address these questions, I investigate explicit and implicit biases among heritage and L2 learners and teachers with the goal of determining if any group holds more favorable bias towards US Spanish, which would signal a greater acceptance of variation, or if they continue to uphold deficit perspectives.

## 5. The Present Study

Participants completed all sections of the experiment in Qualtrics (Qualtrics, Provo, UT), an online survey platform, with IAT integration using Iatgen (Carpenter et al. 2019). They began with a short demographic survey (see Section 5.2) and were able to complete the experiment in 20 to 30 min. Participants were also given breaks in between experiment blocks.

### 5.1. Participant Recruitment

Eighty-one participants that qualified into one of the following groups completed the experiment: students who had taken either heritage or L2 Spanish classes and teachers of Spanish. Heritage learners were either born in the US or immigrated to the US by the age of five and had taken at least 1 year of heritage Spanish language education. L2 learners began learning after the age of 14 and had taken at least 2 years (four semesters) of Spanish as a second language in the US. Teachers[4] based in the US of any level were invited to participate, including primary (*n* = 3), secondary (*n* = 10), and postsecondary (*n* = 10). Participants were recruited via the crowdsourcing site Prolific (www.prolific.co) and through word of mouth (particularly with the teacher group, in which a limited number of Prolific users fit the requirements). The number of participants and demographic information are found in Table 1.

**Table 1.** Participants' demographic information and language background.

| *n* = 81 | Gender Identification | Age Range | Birthplace | L1 Spanish | Mean Years of Spanish Language Learning in School |
|---|---|---|---|---|---|
| Heritage *n* = 28 | Female: 16 Male: 9 Nonbinary: 2 Gender fluid: 1 | 18–25 = 16 26–35 = 9 36–45 = 2 46–55 = 1 | Argentina: 1 Cuba: 1 Guatemala: 2 Mexico: 1 Puerto Rico: 1 USA: 22 | Yes: 28 No: 0 | 7.2 |
| L2 *n* = 30 | Female: 19 Male: 8 Nonbinary: 2 Fluid feminine: 1 | 18–25 = 13 26–35 = 6 36–45 = 6 46–55 = 2 56–60 = 3 | Romania: 1 South Korea: 1 USA: 28 | Yes: 0 No: 30 | 6.2 |
| Teacher *n* = 23 | Female: 17 Male: 6 | 18–25 = 1 26–35 = 8 36–45 = 8 46–55 = 5 56–60 = 1 | Italy: 1 Mexico: 1 Peru: 1 Spain: 2 USA: 18 | Yes: 11 No: 12 | 7.6 |

### 5.2. Stimuli and Design

#### 5.2.1. Experiment 1: Matched Guise Test

Ten speakers provided the audio, five male-identifying and five female-identifying. All were college students or recent graduates in their twenties who were born in California or came to California at an early age (before age 10). All learned Spanish in the household and English in early childhood or learned both Spanish and English in the home, and all speak a variety of US Spanish that was most influenced by Mexican or Guatemalan Spanish. Speakers read a short passage giving directions to a familiar person (see Appendix A), either in standardized Spanish (SS) typical of that which a student might learn from a United States Spanish textbook or a US Spanish repertoire (USS). The USS guise passages contained lexical and lexicalized items common to Spanish(es) in contact with American English in the United States. Passages were written by the author, edited by both Northern and Southern Californian US Spanish languagers, and judged by three Californian Spanish languagers. Two individuals who produced the designated guises (one male and one female) read the same passage in both SS and USS, providing four target guises, which were separated using the audio of the other eight speakers, who each read one passage, serving as 'fillers' to distract listeners from the similarity of voices across the designated guises. Speakers were told to read the passages as naturally[5] as possible, which resulted in all using Spanish-like phonology in their renditions (e.g., 'ticket' pronounced similarly to ˈtiket as opposed to ˈtɪkɪt). The audio sequence is visualized in Figure 1, which contains the two guises, one in each language variety (male, positions 3 and 9; female, positions 6 and 12). Audio samples were cleaned (i.e., background noise removed) in Audacity

(V.3.0.0). The audio samples were organized for a single group of judges (Stefanowitsch 2005), meaning that all participants heard and rated all audio samples (four compared guises and eight fillers) in a within-subjects design for a total of 324 observations.

*Ratings towards SS speakers aggregated and compared to USS.*

| Position | 1 | 2 | 3 | 4 | 5 | 6 | 7 | 8 | 9 | 10 | 11 | 12 |
|---|---|---|---|---|---|---|---|---|---|---|---|---|
| Speaker Type | F | F | G | F | F | G | F | F | G | F | F | G |
| Repertoire | SS | USS | USS | SS | USS | SS | USS | USS | SS | SS | SS | USS |
| Gender | F | M | M | M | F | F | M | F | M | M | F | F |
| Passage | E | A | D | C | B | D | E | C | D | B | A | D |

*Ratings towards USS speakers aggregated and compared to SS.*

**Figure 1.** Matched guise sequence for single group of judges. (Position = Audio in ordered sequence; Speaker Type = Filler (F) and Guise (G); Repertoire = Standardized Spanish (SS) and US Spanish (USS); (Speaker) Gender = Female (F) and Male (M); Passage = Story read (A–E; see Appendix A)).

Participants were prompted to evaluate the speakers on the six social scales below. The language varieties under examination were not mentioned explicitly so as to not prime the participants, aside from prompt four, which avoided mentioning a specific country that may elicit other biases. Participants addressed how much they agreed or disagreed with each statement using a six-point Likert scale[6] (see Figure 2). The prompts are as follows:

1. This person speaks fluently.
2. This person has not fully acquired their language.
3. This person learned their language not only through speaking, but also through reading and writing.
4. This person could communicate easily in a Spanish speaking country.
5. This person is still learning their language.
6. This person would be able to use their language in a professional environment.

S4.Q4

This person learned Spanish not only through speaking, but also through reading and writing.

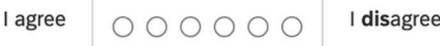

I agree ○ ○ ○ ○ ○ ○ I **disagree**

**Figure 2.** Sample matched guise test question with a six-point Likert scale.

5.2.2. Study 2: Implicit Association Test

Two IAT tests that utilized written labels were issued to participants. The concepts must be relatable to participants completing the experiment; with this in mind, SS is labeled as Spanish and USS is labeled as Spanglish. The first IAT test assessed the strength of associations between the concepts of Spanish and Spanglish with the set of attributes Good/Bad; the second contains the same concepts with the attributes of Academic/Not Academic. The concepts are subjected to named language practices for two reasons: (1) laypeople adhere to the separateness of languages in explicit understanding and are less likely to comprehend what a 'standardized' or 'nonstandardized' repertoire means in cognitive and social terms, and (2) the cognitive load in IAT tests must be reduced for participants to complete the task in a timely manner. The exemplars chosen for Spanish/Spanglish are high-frequency words and typical of a standardized Spanish repertoire taught in a US language classroom and of US Spanish varieties in California, respectively. The attributes present positive/negative binaries, with five exemplars in each category (see Table 2). The chosen exemplars are those

often associated with 'complete' forms of communication (i.e., Good/Academic) and those associated with 'incomplete' or 'broken' forms of communication (i.e., Bad/Not Academic).

**Table 2.** Concepts, attributes, and respective exemplars for both IAT experiments.

| IAT# | Names | Exemplars |
|---|---|---|
| #1, #2 | Spanish | *sin embargo* ('however'), *la camioneta* ('truck'), *el almuerzo* ('lunch'), *alquilar* ('to rent'), *las facturas* ('bills') |
| #1, #2 | Spanglish | *pero like* ('however'), *la troca* ('truck'), *el lonche* ('lunch'), *rentar* ('to rent'), *los biles* ('bills') |
| #1 | Academic | books, tests, school, formal, scholar |
| #1 | Not Academic | slang, informal, street, uneducated, illiterate |
| #2 | Complete | entire, full, intact, whole, perfect |
| #2 | Not Complete | deficient, lacking, fragmented, imperfect, partial |

Both IATs are composed of seven blocks. Blocks 1, 2, and 5 are practice trials that consist of sorting the exemplars with their concept (1 and 5) or attribute (2) head to get acquainted with the terms. Trials 3 and 4 pair Spanish with the 'positive' attribute (Spanish + Good/Academic) and Spanglish with the 'negative' attributes (Spanglish + Bad/Not Academic). In accordance with best practices attested in IAT tests (Lane et al. 2007), concept labels are positioned in the upper left- or right-hand corner of the frame with the attribute below, with the exemplar presented in the center of the screen (see Figure 3).

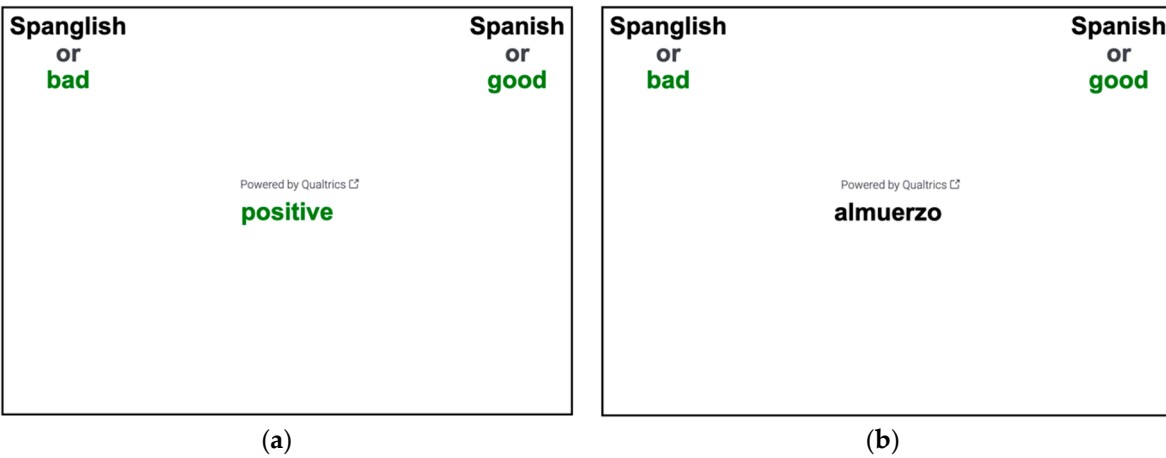

(**a**)                                    (**b**)

**Figure 3.** Screenshots from IAT #1. (**a**) Screenshot from Block 3, test trial, sorting positive and negative attributes to Spanglish + Bad and Spanish + Good. (**b**) Screenshot from Block 4, test trial, sorting language exemplars to Spanglish + Bad and Spanish + Good.

### 5.3. Statistical Models

The responses from the matched guise test were converted into numerical values, meaning that a positive evaluation receives a rating of '6' (e.g., *this person has fully acquired their language*) and negative evaluation '1' (e.g., *this person has not fully acquired their language*), and these ratings were centered. Raw data were then submitted to an exploratory factor analysis (EFA; Helms 2020), a model employed to reveal latent factors by examining the underlying correlations among scales. EFA helps to reduce the number of correlated measures to a set of latent social variables, which serve as the dependent variables in the regression models. A loading value of >0.4 indicates that a variable 'belongs' to the factor and is reported as such (Weatherholtz et al. 2014).[7]

The data were then submitted to mixed-effects linear regression models using the *lmerTest* package (Kuznetsova et al. 2017) in R (R Core Team 2018) with the predictors of

the participant group (heritage; L2; teacher), the guise variety (SS; USS)[8], and potential interactions among the two independent variables. An individual participant was included as a random effect.

All IAT score information was calculated using Iatgen (Carpenter et al. 2019). Participants' response latencies provided in the two IATs were converted into *D* scores (Greenwald et al. 2003; Lane et al. 2007), a measure of the within-subject difference between the compatible and incompatible block means, divided by a pooled standard deviation. Each participant's *D* score represents the subtle differences in effect size. *D* scores range from $-2.0$ to $2.0$, whereby zero represents no difference in response latencies between conditions, a positive score indicates bias towards the expected 'compatible' pairing (i.e., Spanish + Good/Academic), and a negative score signifies bias towards the 'incompatible' pairing (i.e., Spanglish + Good/Academic).

Correlation analyses of the ratings of each grouped dependent variable from the MGT and results from each IAT were carried out in R using the Pearson correlation formula to determine whether linear relationships exist between the implicit bias measures derived from both tests.

## 6. Results

### 6.1. Eliciting Attitudes from the MGT

The raw MGT data submitted to EFA provided two salient factors as determined using Cattell's scree plot. Four evaluative scales (Q1, Q2, Q4, and Q5) grouped together into Factor 1, which pattern together under perceived *acquisition* of the language variety by the speaker. Two scales (Q3 and Q6) are grouped into Factor 2, as they patterned similarly under the perceived attribute of *academic-ness* of the language learned and produced (i.e., learned the language in school); see Table 3.

**Table 3.** Loadings of rating scales for guises in EFA on Factor 1, *acquisition*, and Factor 2, *academic-ness*. Loadings above an absolute value of 0.4 are bolded.

| Rating Scale | Factor 1: *Acquisition* | Factor 2: *Academic-Ness* |
|---|---|---|
| Q1. Is fluent | **0.7** | 0.49 |
| Q2. Incomplete acquisition | **0.84** | 0.26 |
| Q3. Learned by speaking, reading, and writing | 0.22 | **0.55** |
| Q4. Speaks 'globally understood' language | **0.6** | 0.51 |
| Q5. Done learning language | **0.59** | 0.24 |
| Q6. Learned language in school | 0.56 | **0.77** |

The regression model fit onto the factor *acquisition* demonstrates a main effect of the guise variety, and all participant groups rated SS more positively than USS ($p < 0.0001$), meaning that when speakers employed SS, they were perceived to have acquired their language more fully than when they employed a USS repertoire (see Table 4 and Figure 4).

**Table 4.** Summary of Mixed Effects Linear Regression Model Fitted to perceived *acquisition*.

| | $\beta$ **Coefficient** | **Standard Error** | *t* | *p* **Value** |
|---|---|---|---|---|
| (Intercept) | $-0.592$ | 0.111 | 5.314 | 0.0001 |
| Participant—L2 | 0.322 | 0.155 | 2.082 | 0.03 |
| Participant—Teacher | 0.229 | 0.166 | 1.381 | 0.2 |
| Guise Language—SS | 0.885 | 0.127 | 6.974 | 0.0001 |

The intercept for this model is heritage participants evaluating USS speakers. Positive $\beta$ values indicate that the participant evaluated the speaker as demonstrating more complete acquisition of their variety. The estimated variance of the random effect of the listener is 0.122.

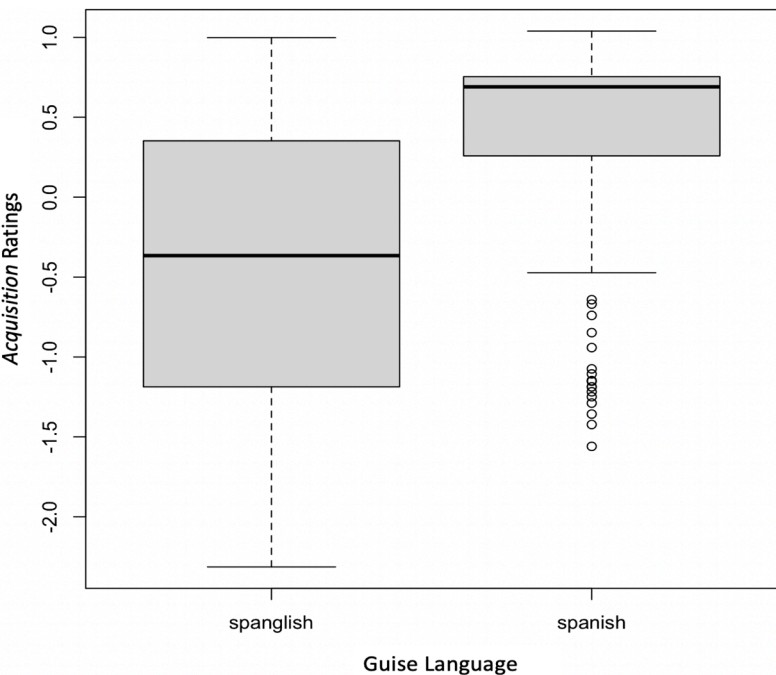

**Figure 4.** Boxplot showing listener evaluations of speakers' perceived *acquisition*, conditioned by the guise variety.

The regression model fit onto the *academic-ness* factor (see Table 5) once again demonstrates a significant main effect of the guise variety ($p < 0.0001$), as all participant groups rate USS less favorably than SS on the basis of perceived *academic-ness* of the speaker's learned variety (see Figure 5).

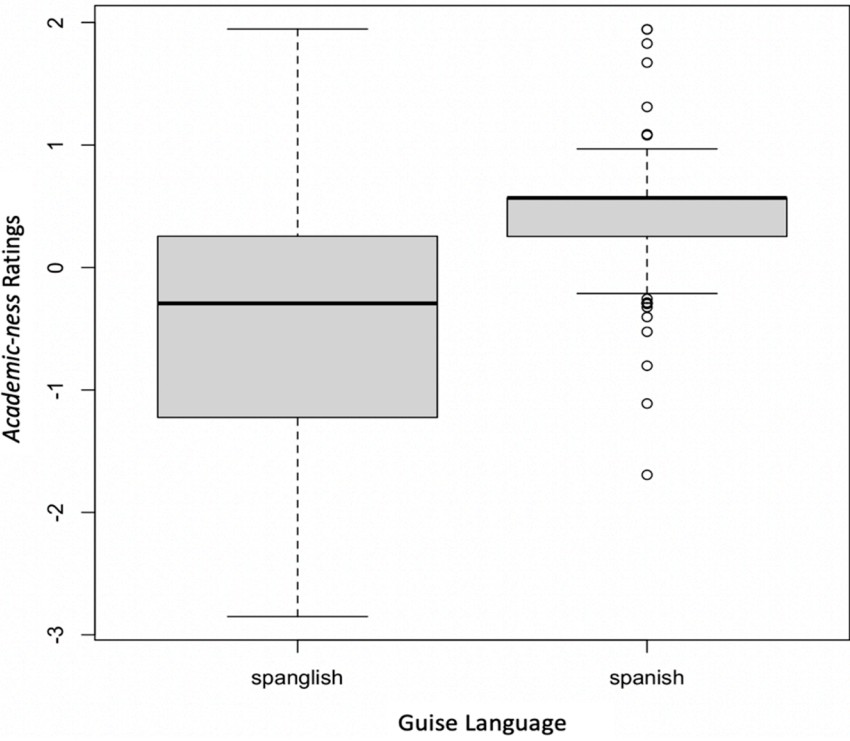

**Figure 5.** Boxplot showing listener evaluations of speakers' perceived *academic-ness*, conditioned by the guise variety.

**Table 5.** Summary of Mixed Effects Linear Regression Model Fitted to perceived *academic-ness*.

|  | *β* **Coefficient** | **Standard Error** | *t* | *p* **Value** |
|---|---|---|---|---|
| (Intercept) | 0.591 | 0.1052 | −5.615 | 0.0001 |
| Participant—L2 | 0.213 | 0.146 | 1.454 | 0.148 |
| Participant—Teacher | 0.287 | 0.157 | 1.834 | 0.06 |
| Guise Language—SS | 1.0241 | 0.121 | 8.48 | 0.0001 |

The intercept for this model is heritage participants evaluating USS speakers. Positive *β* values indicate that the participant evaluated the speaker's variety as having been acquired in an academic setting. The estimated variance of the random effect of the listener is 0.116.

### 6.2. Assessing Implicit Bias

In the Spanish/Spanglish + Good/Bad IAT (see Table 6), positive *D* scores indicate that all participant groups associate Spanish more strongly with 'Good' than Spanglish due to their faster reaction times when categorizing exemplars to Spanish + Good than to Spanglish + Good (see *p* values). The timeout rate was low at <0.008%, the drop rate (number of participants dropped for overly fast responding under 300 ms) was zero for the heritage and L2 groups and one for teachers, and the error rate (percent of trials that were incorrect) was under 0.09%.[9] A one-way ANOVA modeled to these results suggests that the L2 group had significantly lower *D* score means than the heritage and teacher groups.

**Table 6.** Spanish/Spanglish + Good/Bad IAT data information for all participant groups.

| **Participants,** *n* | *D* **Score Mean** | *t***-Test** | *p***-Value** |
|---|---|---|---|
| Heritage, 28 | 0.51 | 10.39 | 0.00001 |
| L2, 30 | 0.35 | 4.1 | 0.0003 |
| Teacher, 22 | 0.49 | 4.32 | 0.0003 |

In the Spanish/Spanglish + Academic/Not Academic IAT (see Table 7), positive scores determine that participants associate Spanish more strongly with Academic due to their faster reaction times when the two were paired. These reaction times differed significantly from the reaction times when Spanglish was paired with Academic (see *p* values). The timeout rate was low at <0.002% and the drop rate was zero for the heritage and L2 groups and one for teachers. The error rate was low at 0.08%. A one-way ANOVA once again demonstrates that the L2 group had significantly lower *D* score means than the heritage group.

**Table 7.** Spanish/Spanglish + Academic/Not Academic IAT data information for all participant groups.

| **Participants,** *n* | *D***-Score Mean** | *t***-Test** | *p***-Value** |
|---|---|---|---|
| Heritage, 28 | 0.63 | 9.43 | 0.00001 |
| L2, 30 | 0.54 | 7.29 | 0.00001 |
| Teacher, 22 | 0.58 | 7.27 | 0.00001 |

### 6.3. Correlation Analyses: MGT to IAT

Correlation analyses were carried out to determine the relationship of MGT ratings to IAT results. Among the three listener groups (heritage, L2, and teacher), each MGT result set (*acquisition* and *academic-ness*) was separated by the guise variety (SS and USS), which were each paired with each IAT result set. All models demonstrated weak and/or non-significant relationships between explicit attitudes elicited indirectly from the MGT and automatic bias elicited from the IAT. Values closer to −1 indicate a strong negative correlation and values closer to +1 a strong positive correlation. Correlations below +/−0.4 are considered weak using social science standards, even in cases in which they are presented with *p* values less than 0.05 (Evans 1996).

However, there are some correlations approaching practical significance (>0.4) that should be discussed. Both the L2 and teacher groups have a moderately weak negative correlation (−0.36 and −0.34, respectively) between their Spanish + Academic IAT scores and MGT *academic-ness* scores for USS speakers. That is, as IAT scores increased (favoring Spanish, not Spanglish), *academic-ness* evaluations of USS speakers went down (favoring SS). Also, both the teacher and heritage groups have a moderately weak positive correlation (both 0.37) between their Spanish + Academic IAT scores and MGT *acquisition* evaluations for SS speakers, meaning that as IAT scores increased (favoring Spanish), *acquisition* evaluations of SS speakers increased as well (favoring SS). Interestingly, the L2 group also demonstrated a moderately weak negative correlation of Spanish + Academic (IAT) with the MGT *acquisition* evaluations for SS speakers (−0.36). This indicates that IAT scores went down as MGT scores went up, meaning that while they increasingly favored SS over USS speakers, their positive implicit bias of Spanish + Academic weakened (i.e., they had slower reaction times in their associations of the two). This is evidenced in their lower IAT scores of the three groups (see Table 8).

**Table 8.** Correlation analyses of MGT (separated by language) to IAT results, presented by group.

| | | Spanish | | US Spanish | |
|---|---|---|---|---|---|
| | *Results* | Acquisition (MGT) | Academic-ness (MGT) | Acquisition (MGT) | Academic-ness (MGT) |
| Heritage learners | Spanish + Good (IAT) | $r = 0.37$ $p < 0.004$ | $r = 0.16$ $p < 0.25$ | $r = -0.13$ $p < 0.34$ | $r = -0.02$ $p < 0.86$ |
| | Spanish + Academic (IAT) | $r = 0.13$ $p < 0.34$ | $r = 0.8$ $p < -0.03$ | $r = -0.03$ $p < 0.83$ | $r = -0.16$ $p < 0.25$ |
| L2 learners | Spanish + Good (IAT) | $r = 0.09$ $p < 0.5$ | $r = -0.06$ $p < 0.62$ | $r = 0.18$ $p < 0.15$ | $r = -0.36$ $p < 0.003$ |
| | Spanish + Academic (IAT) | $r = 0.11$ $p < 0.41$ | $r = -0.36$ $p < 0.005$ | $r = -0.06$ $p < 0.66$ | $r = -0.06$ $p < 0.62$ |
| Teachers | Spanish + Good (IAT) | $r = 0.05$ $p < 0.75$ | $r = -0.06$ $p < 0.69$ | $r = -0.23$ $p < 0.12$ | $r = -0.17$ $p < 0.27$ |
| | Spanish + Academic (IAT) | $r = 0.37$ $p < 0.01$ | $r = 0.21$ $p < 0.18$ | $r = -0.25$ $p < 0.09$ | $r = -0.34$ $p < 0.02$ |

## 7. Discussion

This study is the first to employ both the MGT and IAT to examine the effects of Spanish language training (i.e., heritage, L2, and teacher) and bias on perceptions of both standardized Spanish and US Spanish repertoires in the United States. The MGT was used as an indirect method to gauge bias and provide insight into how lexical items index differentiated social meanings depending on the repertoire presented. The dependent variables, perceived *acquisition* and *academic-ness* of the speaker's variety, were heavily conditioned by the guise variety, as SS was rated more positively in each attribute with no listener profile group differences. With regard to *acquisition*, these findings demonstrate that lexical items of nonstandardized or nonhegemonic language in the case of US Spanish index ideologies of incomplete acquisition that ignore the dynamic possibilities of multilingual languagers. This steadfast bias is rooted in the separateness of languages as inherent to the alleged learning, storage, and production of language. In specific relation to language variation research, this lends new insights into the indexical field of US Spanish and its salient lexical features, an investigation that should be replicated and expanded with other variables of US Spanish (see Román et al. 2019). With regard to the factor of *academic-ness*, once again all participants evaluated the SS speakers as having learned their variety in an academic setting. These results associate USS with non-academic settings for learning languages, such as the home or community, but also might indicate a conscious knowledge

that USS is not 'formally' learned in the classroom, which is in line with how heritage language programs generally promote the acquisition of an 'academic' register. Similarly, these perspectives are highly affected by monoglossic influences, whereby assimilation to SS in an academic setting is normalized and expected. Thus, SS and USS seem to be mutually exclusive codes that index different social meanings when employed by the same person. As participants demonstrated in their MGT evaluations, USS speakers are significantly less likely to have learned their repertoires in school, as 'academic' language in this case lines up neatly with the boundaries of named, codified languages. As such, the use of USS in the classroom appears incongruent with academic language use.

In the assessment of associations between the concepts of Spanish/Spanglish and the attributes of Good/Bad and Academic/Not Academic, all participant groups demonstrated faster reaction times when Spanish was paired with the positive qualities. These findings overwhelmingly indicate that participants more strongly associate Spanish with the positively charged exemplars related to Good and Academic. These subconscious biases are affirmed in the MGT, where ideologies of 'complete' *acquisition* and *academic-ness* in language variety expression are again associated with SS. As indicated in the ANOVA model, heritage participants had stronger positive bias towards Spanish + Good and Spanish + Academic (i.e., faster reaction times in associating the positive attributes to Spanish over Spanglish) than the L2 group, and teacher averages patterned similarly to the heritage group. Though all groups still demonstrated more positive bias to Spanish over Spanglish, L2s' lower averages for Spanish + Good may indicate that students do not learn these associations with Spanish repertoires as early or as directly as the heritage group, who have been exposed to Spanish and have developed metalinguistic awareness of their own Spanish(es) since an early age. "Likewise, the difference between L2 students' averages for Spanish + Good and those of teachers approached significance ($p = 0.06$), suggesting that teachers pattern similarly to the heritage group. This could be due to the teachers' exposure to sociolinguistic ideologies in school, even in the absence of the home exposure that characterizes the experience of heritage students.". These results highlight the expectations that SS, or the named language 'Spanish,' is a more 'complete' and 'whole' language affiliated with an academic realm. Heritage students hold these biases and may also learn them early in life within the community and family, L2 learners develop them further in Spanish language study, and teachers may implicitly (or explicitly) promote them in their teaching. A deeper dive into the experiences of heritage languagers as students *and* heritage teachers can shed light on how their metalinguistic awareness of the indexical meanings associated with USS is cultivated in youth (see Ek et al. 2013).

This study also attempted to reveal if data trends from each MGT variable (i.e., *acquisition* and *academic-ness*) correlated with each IAT (Spanish + Good/Spanish + Academic). Results presented generally weak correlations; however, there were some significant correlations that approached practical significance. It has been attested that the associations that participants make in the IAT are less susceptible to change (Petty et al. 2009; Evans 2008; Karpen et al. 2012), and that explicit attitudes are more apt to change with influence. Thus divergences of more malleable attitudes are possible with social and systemic change.

In light of the results discussed above, the research questions can now be addressed. The first research question asked if US Spanish lexical features were salient to L2 and heritage learners as well as teachers of Spanish. Findings revealed that US lexical items are indeed salient to all groups. This was demonstrated in the MGT evaluations, wherein Spanglish speakers were rated lower on both social attributes (*acquisition* and *academic-ness*). Similarly, the results of L2 learners also display that, even if this group had less exposure to SS ideologies in their early learning, they have learned enough about stigmatized lexical features of US Spanish to be able to identify them in the audio and rate the USS speakers less favorably. In response to the second research question, which asked to what extent differential participation in language education programs (L2 students, heritage students, and teachers) correlates with more positive explicit attitudes toward US Spanish features, the statistical analysis found no clear correlation between program type and more positive

attitudes. These findings demonstrate that L2 and heritage learners as well as teachers of Spanish exhibit similarly patterned bias across testing conditions, which suggests that heritage language education, whether it seeks to value translanguaging or not, is not associated with a change to heritage languagers' malleable attitudes (elicited from the MGT) from deeply rooted bias (from the IAT).

Finally, the third research question posed whether the attitudes and associations elicited from indirect methods (i.e., the MGT) and automatic response (i.e., the IAT) suggest that explicit and implicit biases result from distinct cognitive processes, in line with previous research examining explicit to implicit bias. Results pattern similarly to previous findings comparing explicit to implicit bias ratings, whereby correlations were weak to moderately weak, indicating that the processing of varying biases—even if they present similar findings (e.g., US Spanish or Spanglish is seen as less academic)—is complex and not monolithic, contributing insights into theories that examine implicit and explicit attitudes as structurally distinct (Greenwald and Nosek 2009). Examinations of distinct biases garnered from attitude measures like the MGT and the IAT are "able to capture distinct levels of linguistic attitudes which are potentially conflicting" (McKenzie and Carrie 2018, p. 837); however, in the case of the experimental line presented here, results are stable across measures (i.e., favor SS/Spanish over USS/Spanglish) even if they do not correlate. The biases derived from the MGT and the IAT, while they do not strongly correlate in this study, are also not diverging with regard to longstanding deficit perspectives of US Spanish. While the results of this study do not support the IED hypothesis, the demonstration of weak or nonexistent correlations between the bias measures contributes to the previous language attitude literature that has examined explicit and implicit bias as different processes (see McKenzie and Carrie 2018; Ianos et al. 2020).

As this study demonstrates, as well as the literature evaluating raciolinguistic ideologies in schools, US Spanish speakers who navigate their linguistic repertoires easily and effectively are still thought to index linguistic deficiency by their perceiving subjects, who are composed in this study of both L1 and L2 Spanish languagers, demonstrating that raciolinguistic ideologies pathologize language users. Despite the increase in heritage language and bilingual programs in the US in recent decades, standard language and monoglossic ideologies continue to dominate the ways in which language education in the United States is conceptualized and formalized through early teachings of who is a 'valid' languager in an academic setting. As mentioned previously, these ideologies rely heavily on the naming conventions of idealized languages that undergo standardization, exhibiting a perhaps subconscious inability for all groups to view speakers of languages 'in between' as fluent and competent in their communication. Relatedly, participants' perceptions delegitimize translanguagers in the US and beyond, a position that is undergirded by both folk ideologies and empirical research that maps the political boundaries of named languages onto real language production, disparaging language change and variation, which are inherent to all language communities. It has even been posited that the term 'Spanglish' somewhat iconicizes (see Gal and Irvine 2019) deficits or symbolizes such disparities (see Otheguy and Stern 2011), particularly within an academic context, as the very term visibly indicates hybridity or 'incompleteness' within the bounds of one language. These deficit perspectives contribute to the raciolinguistic ideologies that plague how racialized students are perceived, how they are educated in the classroom, and their trajectory in the US education system. This status quo leads multilingual Latinx students to experience *languagelessness*, or the incapability of producing a legitimate language according to hegemonic standards (Rosa 2016), leading them to view their own repertoires as insufficient for an academic context. As mentioned earlier, there is no consensus as to what 'academic' language is (Flores 2020a; Martínez and Mejía 2020; Valdés 2004). These biases must be discussed in conjunction with the so-called "word gap" literature that targets racialized children early on in their education, disenfranchising them from equitable learning experiences that should center their dynamic multilingual practices as natural language development and expression.

## 8. Conclusions

This study shows that L2 and heritage students as well as teachers demonstrate stable explicit attitudes (elicited indirectly in the MGT) and implicit attitudes (elicited from the IAT) towards indexical meanings linked with US Spanish lexical features. Likewise, results exhibit that varying implicit biases may not be stored and processed in the same place and manner, providing hope that positive changes in how languages are conceptualized and taught can shift semi-implicit attitudes and eliminate discriminatory meanings associated with the indexical fields of USS lexical items. The study, however, presented some limitations that may inspire further exploration of the topics of bias and standard language ideologies. First, the design of the study presents the two language varieties in a dichotomous fashion, which is not how language and language communities are encountered in real life. Also, the stimuli presented in the MGT were audio while those in the IAT were written. A comparison of similar stimuli or replication of this study design would shed light on these effects. Relatedly, the Spanish phonologically inflected lexical items in the USS guise passages index particular and unique indexical meanings, which can shift and change with differing stimuli; in other words, if this study had focused on morphosyntactic variation or the inclusion of phonetic features associated with Californian English, the salience of variables may have shifted as well as the indexical field of the variables under examination. The order of guises can also affect perception; that is, hearing the male USS guise before the same male SS guise may affect how the second is heard (and same for the female guise in both varieties). Lastly, a larger participant pool may strengthen the analysis and highlight the potential interactions among independent variables.

I encourage scholars engaging with this topic to accept the possibilities of language liberation and to reject discriminatory linguistic policies and practices that continue to racialize and exceptionalize multilingual languagers as in need of remediation. Many people are already doing vital work to shift the harmful associations with US Spanish(es) through the reframing of Spanish language education. Prada (2019), Flores (2020b), and Seltzer and Wassell (2022) demonstrate how translanguaging allows for oppressive structures in the classroom to be reconfigured and reimagined to center students' natural language expression. De Los De Los Ríos et al. (2021) explore the connection of cultural practices and translingualism in fostering language exploration and identity expression. Train (2020) promotes the teaching of variation as a means to achieve social justice, Holguín Holguín Mendoza (2018) privileges the home and community repertoires in the planning of a heritage language program, and Bucholtz et al. (2018) explore the empowerment of Latinx youth through community language work and participatory action research. Anya (2021) demonstrates the benefits of critical race pedagogy for more inclusive world language education, which Austin (2022) takes a step further by countering anti-Black racism in teacher training and instilling reflexivity. Relatedly, (Licata et al. Forthcoming) exhibit the benefits of a course on raciolinguistic theory and practice to increase metalinguistic awareness in both language teachers and students. There are many more examples of people laying this vital groundwork, and we must interrogate how widespread implementation of such curricula across language programs and teacher training can shift perceptions of US Spanish(es) in academic contexts via the collective rejection of deficit perspectives towards US Spanish(es) and Latinx students. Only then can we reimagine safer and more affirmative learning experiences for our heritage learners, as well as foster the development of empathy among teachers and L2 students.

**Funding:** This research was funded by the Romance Languages and Literatures degree program at UC Berkeley.

**Institutional Review Board Statement:** The study was conducted in accordance with the Declaration of Helsinki and approved by the Institutional Review Board of the University of California, Berkeley (protocol code: 2018-04-10957, approved on 24 September 2020) for studies involving humans.

**Informed Consent Statement:** Informed consent was obtained from all subjects involved in the study.

**Data Availability Statement:** Not applicable.

**Acknowledgments:** The author expresses gratitude for the support and advising provided by Justin Davidson and Keith Johnson (UC Berkeley), Kathryn Campbell-Kibler (Ohio State University), and Annie Helms and Ernesto Topete Gutiérrez (UC Berkeley) on this project. Many thanks to the several participants who provided audio, consulting, or completed the experiment. She also thanks undergraduate research assistants Damaris Velázquez, Marcos Lobato Scharfhausen, and Jesus Duarte for piloting the experiment. Special thanks to the Department of Romance Languages and Literatures at UC Berkeley for funding this study. Lastly, this paper is dedicated to Jon Henner.

**Conflicts of Interest:** The author declares no conflict of interest.

**Appendix A. Guise Stories (English Words Bolded Refer to SS and Those <u>Underlined</u> Refer to USS)**

|  | Standardized Spanish (SS) Repertoire | US Spanish (USS) Repertoire | English Translation |
|---|---|---|---|
| Story A | **Bueno**, para ir al **muelle**, tienes que doblar a la derecha en la **calle** Retiro. Puedes dar vuelta en Pacheco, pero hay mucho **tráfico** ahí **todo el tiempo**. **Ya sabes**, es mejor evitar **los semáforos**. Te espero en **la camioneta verde** de mi papá. **Bueno**, nos vemos ahí. | <u>So</u>, para ir al <u>pier</u>, tienes que doblar a la derecha en <u>Retiro Street</u>. Puedes dar vuelta en Pacheco, pero hay mucho <u>traffic</u> ahí <u>all the time</u>. <u>You know</u>, es mejor evitar las <u>traffic lights</u>. Te espero en <u>la troca verde</u> de mi papá. <u>Ok pues</u>, nos vemos ahí. | **Well**/<u>So</u>, to go to the pier, you have to turn right on Retiro Street. You can turn on Pacheco, but there is a lot of traffic there all the time. You know, it's better to avoid the traffic lights. I'll be waiting for you in my dad's green truck. Ok, see you then. |
| Story B | Para llegar a la tienda, tienes que **tomar la carretera** que va al **centro**. **Sin embargo**, habrá mucha gente porque es la hora **del almuerzo**. Todos irán a los restaurantes durante **el descanso** para **comprar comida**. Yo estoy **lleno/a y** no voy a comer, **pero** si quieres, podemos **llegar** al restaurante. | Para llegar a la tienda, tienes que subirte al <u>freeway</u> que va al <u>downtown</u>. <u>Pero like</u>, habrá mucha gente porque es la hora <u>del lonche</u>. Todos irán a los restaurantes durante <u>el break</u> para <u>agarrar</u> comida. Yo estoy <u>full so</u> no voy a comer, <u>pero like</u>, si quieres, podemos <u>parar</u> al restaurante. | To arrive at the store, you need to get on the freeway and go downtown. **However**/<u>However</u> or <u>But like</u>, there will be a lot of people because it's lunch time. Everyone will be going to restaurants during break to get food. I'm full so I won't be eating, however/<u>but like</u>, if you want, we can stop at the restaurant. |
| Story C | **Pues** para llegar al parque, debes **dar vuelta a la izquierda** en **la calle** Olivares. Puedes dejar el carro en **la estructura de estacionamiento** de la esquina, pero no sé si **está abierta**. **De hecho**, mejor pasa por la calle San Andrés y por ahí puedes entrar. Si prefieres, podemos hacer las compras para **la fiesta** en la tienda cerca **de la parada**. También, tengo que **devolver** unas cosas que ya no me sirven. | <u>So</u>, para llegar al parque, debes hacer una izquierda en Olivares <u>Street</u>. Puedes dejar el carro en <u>el parking lot</u> de la esquina, pero no sé si <u>están abiertos</u>. <u>Actually</u>, mejor pasa por la calle San Andrés y por ahí puedes entrar. Si prefieres, podemos hacer las compras para <u>el party</u> en la tienda cerca <u>del bus</u>. También, tengo que <u>regresar</u> unas cosas que ya no me sirven. | So, to get to the park, you need to turn left on Olivares Street. You can leave your car in the parking lot on the corner, but I don't know if they are open. Actually, it's better to pass through San Andrés Street and you can enter through there. If you want, we can shop for the party in the store near the bus stop. Also, I have to return some things that I don't need anymore. |
| Story D | **Bueno**, la **película** empieza a las ocho. Si quieres comer antes, podemos ir al **restaurante** que está cerca. **Pero**, Daniel no puede entrar porque hay un bar y todavía está en **la escuela secundaria**. Nos reunimos enfrente del **supermercado**. Primero voy **de compras** con Elena, quien también quiere **platicar** sobre los planes para este **fin de semana**. Una cosa más: ¡no te olvides de los **boletos de entrada**! | <u>So</u>, la <u>muvi</u> empieza a las ocho. Si quieres comer antes, podemos ir al <u>restaurán</u> que está cerca. <u>Pero like</u>, Daniel no puede entrar porque hay un bar y todavía está en <u>la high school</u>. Nos reunimos enfrente de la <u>marketa</u>. Primero me voy <u>shopping</u> con Elena, quien también quiere <u>discutir</u> sobre los planes para este <u>weekend</u>. Una cosa más: ¡no te olvides de los <u>tickets</u>! | **Well**/<u>So</u>, the movie starts at eight. If you want to eat before, we can go to the restaurant that is nearby. **But**/<u>However</u> or <u>But like</u>, Daniel can't go in because there's a bar and he's still in high school. Let's meet in front of the market. First, I'll go shopping with Elena, who also wants to discuss plans for this weekend. One more thing: don't forget the tickets! |

| | | | |
|---|---|---|---|
| Story E | **Pues**, la ruta más rápida es por la **Avenida** Paloma. Pero quizás **esté** cerrada, **así que** puedes también pasar por la calle Francisco. Aunque tal vez **llegues** tarde—**ya ves**, siempre hay mucho tráfico y poco **estacionamiento**. Javier nos va a acompañar porque **renunció** a su trabajo y ya no tiene que trabajar **por** las noches. Cuando estés **listo**, ¡**envíame un mensaje**! | So <u>pues</u>, la ruta más rápida es por <u>Paloma</u> <u>Avenue</u>. Pero quizás <u>estará</u> cerrada, **so** puedes también pasar por la calle Francisco. Aunque tal vez <u>estés</u> tarde—<u>you know</u>, siempre hay mucho tráfico y poco <u>parkin</u>. Javier nos va a acompañar porque <u>cuitió</u> su trabajo y ya no tiene que trabajar <u>en</u> las noches. Cuando estés <u>ready</u>, ¡<u>textéame</u>! | **So/**<u>So like</u>, the fastest route is down Paloma Avenue. But maybe it will already be closed, so you can also go down Francisco Street. Even if you get there late—you know, there's always a lot of traffic and little parking. Javier is going to accompany us because he quit his job and now, he doesn't have to work at night. When you are ready, text me! |

## Notes

[1] In this paper, and more broadly, I utilize *languager* to generally indicate any person that communicates and perceives language. I use *speaker* when the form of communication is specifically related to oral production and hearing.

[2] In line with how Spanish language programs are often organized, heritage language learning encompasses classes for languagers who are raised in a home where a non-hegemonic language is used, and who can use or comprehend the home language to some degree, and who are, to any degree, bilingual in that language and in the hegemonic variety (Valdé s et al. 1999). Second language (L2) learning refers to courses in which languagers are learning or have learned an additional language in school. With regard to this study, L2 learners began learning Spanish in school after the age of 14.

[3] The generalized idealized communicator is a hearing subject (see Henner and Robinson 2021), thus colonial epistemologies also take phonocentric, ableist stances.

[4] L1 variability within the teacher group did not significantly mediate bias differences.

[5] See Tamminga (2017), who found no evidence that social evaluations of /iŋ/~/ɪn/ variation differed across frame utterance styles.

[6] A six-point scale has been shown to increase discrimination and reliability than a five-point scale (Chomeya 2010).

[7] An exploratory factor analysis is an unsupervised language model that requires human expertise in the analysis of factors. As some evaluative scales may score a loading factor above 0.4 for more than one factor, researcher judgment ultimately determines factor grouping.

[8] The only statistical comparisons of relevance to my research questions are those concerned with whether or not each group of listeners differentiated each of the target language guises (SS/USS) by the attitudinal category, namely a main effect of the guise variety or an interaction effect between the guise variety and listener profile group. Accordingly, I limit my reporting and discussion of results to these effects and potential interactions. Any potential main effect of the listener profile group would reflect non-substantive comparisons of averaged *acquisition* or *academic-ness* ratings by group (i.e., there is no meaningful interpretation of one participant group having higher ratings over other groups unless those higher ratings interact with differentiated ratings mediated by the guise variety).

[9] Errors were replaced with participant block means of correct trials plus 600 ms (or the D600 procedure; Greenwald et al. 2003).

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
