# Peer review of "Indexing Deficiency: Connecting Language Learning and Teaching to Evaluations of US Spanish"

_languages, doi:10.3390/languages8030204_

Round 1
Reviewer 1 Report
This paper utilizes Matched Guise Technique and IAT to measure the explicit and implicit biases of three groups of listeners - heritage learners, L2 learners, and teachers of Spanish - towards "standard" and US Spanish lexical items. The use of these two techniques together to examine attitudes towards US Spanish is novel and thus a valuable contribution to the field. The design and statistical analysis are sound.
Specific comments are to be found in the document itself.
The linguistic and experimental elements of the paper need better support. Currently, the article repeats the same information about ideologies (and relies too heavily on Rosa & Flores 2017) several times but does not give sufficient discussion to lexical characteristics of US Spanish, literature on codeswitching/translanguaging (other than the stigma these carry, this is already emphasized), characteristics of heritage learners, and the relationship between implicit/explicit judgements (especially in relation to the IAT, which is not as familiar to readers as MGT). Other research of implicit/explicit judgements and how past studies (even outside of linguistics) use these, especially together, is missing. The author does not sufficiently connect their work to the linguistic literature on bias.
The author also needs to be more specific about the limitations of the study. The study is limited to the study of lexical items from US Spanish (which is totally fine, of course not all elements can be studied in a single investigation!), but makes broad claims about the findings. If instead of lexical items, the guises differed in morphosyntax, for example, findings might be different. The author also does not specify if the readers used Spanish-like or English-like phonology in their US Spanish guise, which could influence findings (and might explain differences in gender if the male reader differed in how he read the US Spanish lexical items - that is, if he used English phonology and the female speaker used Spanish phonology).
The gender difference (the male speaker being rated lower than the female speaker in the USS guise) is not sufficiently explored.
More information on the demographic characteristics would be helpful. Heritage speakers in particular are a very diverse group - were there differences in these individuals based on their profiles (like if they were born in the US or came closer to age 10?); were the teachers mostly higher education or elementary (and were there any differences between levels); how advanced were the L2 speakers and had they studied abroad/had lots of US Spanish contact (geographic location in the US would influence the latter!)?
Of the 10 voices, why did the author use the particular speakers and guises for the study?

Reviewer 2 Report
Indexing deficiency: Connecting language learning and teaching to evaluations of US Spanish
I enjoyed reading this interesting and generally well-written manuscript investigating students’ MGT and IAT attitudes towards Standard Spanish and US Spanish.
For me, the manuscript is publishable, and I have a few comments/suggestions which may be of help to the authors – especially regarding the study narrative, operationalisation of implicit attitudes and the detailing of potential study limitations.
Abstract
Very clear, although if space allows, I wonder whether the authors can specify the number of participants, who these participants are, and mention the words evaluations and/or attitudes (the aim of the study)?
Introduction
Para 1: Would it be worth specifying and referencing what is meant by a ‘standardized language’? (Sorry if I have missed)
3. Evaluating implicit social cognition and linguistic bias (and passim throughout the manuscript)
Definitions of explicit vs implicit attitudes/measures – beyond conscious/unconscious – huge debate within psychology, e.g., see Greenwald vs Gawronski research.
Specifically, despite early claims by Greenwald, it is increasingly controversial to label implicit attitudes as unconscious or subconscious (and what’s the difference between un/subconscious?) as stated in this manuscript (and passim). There exists a large body of research in psychology refuting such broad claims. As such, it is perhaps more usual – and safer - to label implicit attitudes in terms of automatic/non-verbalisable.
Likewise, it seems very controversial to assert that the MGT, as an indirect approach, can tap into implicit attitudes. For instance, in a recent in-depth monograph, McKenzie and McNeill (2023: 16-18) differentiate between indirect measures (such as the MGT) where participants may be aware their attitudes are being tested but are not sure of the precise attitudinal object (i.e., language varieties) versus implicit measures (such as the IAT) which can access automatic evaluative responses (i.e., implicit attitudes). From this perspective, both indirect and direct instruments tap into explicit/self-report attitudes and, by contrast, implicit measures tap into automatic attitudes.
In short, if the authors wish to maintain the MGT can tap into implicit attitudes, I recommend a much deeper and more nuanced discussion which takes the current Social Psychology literature into account. Again, the concept of a ‘gradience of implicitness’ is highly controversial and needs to be discussed in depth – beyond Rosseel & Grondelaers, 2019 interesting review paper – with reference to current social psychology literature (the very nature of consciousness is a very difficult topic).
Some good discussion of IED – very interesting and well done. Again, for a much deeper discussion than McKenzie and Carrie’s (2019) pilot study paper – re role of IED in language attitude change - suggest to look at McKenzie and McNeill (2023) reporting of the large-scale study findings (especially p21-22, 39-42)
https://www.routledge.com/Implicit-and-Explicit-Language-Attitudes-Mapping-Linguistic-Prejudice-and/McKenzie-McNeill/p/book/9780367703530
Generally, prior to RQs. The study is extremely interesting and well designed (see below). However, I wonder whether the authors can be more specific re the research niche – i.e., how does the study attempt to extend prior research in the Spanish-speaking context (sorry if I have missed)?
P7, line 346 (ish) – in light of above, suggest that RQ2 i.e., ‘gradience of implicit bias’ is reformulated to include mention of indirect/explicit MGT attitudes and IAT implicit attitudes
Method
For me, the Participants data would be better placed first to help the reader. Given the number of DVs and IVs examined N=81 is quite low, please detail as a potential limitation of the study
Can I also suggest to highlight MGT asstudy 1 and IAT as study 2 – found the placing of instruments together a little confusing (happy to leave authors to decide though)
5.1.1. Matched guise test
For me, would be good to have a short discussion of the advantages and disadvantages of employing read speech – as opposed to natural, spontaneous speech – as stimuli. See Van Bezooijen and Gooskens, 1997 - https://journals.sagepub.com/doi/10.1177/0261927X99018001003 (may be worth detailing as a potential limitation of the study)?
Please also justify the specific traits employed in the evaluative scales and use of 6-point scale
5.1.2. Implicit association test
Similarly, justify choice of exemplar traits in the IAT
6. Results
Some good analysis here – well done
6.1. Assessing covert attitudes from MGT
‘Covert attitudes towards male- and female-identifying speakers of both SS and USS were 472 elicited using the matched guise technique- - why use covert here?
For me, Q4 (and Q6?) loads on both dimensions – detail as a potential limitation?
If possible, please refine table titles a little, I have a background in statistics but have some trouble interpreting the output (clearer table titles will help)
6.3 Correlation analyses: MGT to IAT
This section seems a little weaker. Please detail the p value and correlation co-efficient for each of the correlation analysis undertaken.
As per above (and passim) suggest to reframe ‘All models demonstrated weak and non-significant relationships between more malleable implicit bias (elicited from the MGT) and more unchanging, unconscious bias (elicited from the IAT)’
Could the difference between the ratings obtained by the MGT and IAT be the result of differing stimuli, i.e., read speech vs lexis? As there is no way to test, suggest to detail as a potential limitation (recognising limitations does not invalidate your interesting results)
7. Discussion
Some further examples of implicitness and others which I suggest to reframe:
‘this study also presents the first comparative analysis of implicit bias as a gradient scale to assess how subconscious attitudes relate to one another (or not).’
‘The MGT was used to gauge a range of semi-implicit biases’
‘undergirded by a commonly attested expectation across Western cultures that men employ less standardized variants’ – for me, this is a much too general – see Labov’s well-known and oft cited principles of gendered speech re incoming vs stable variants, so suggest to hedge or delete
8. Conclusions
‘…has maintained early learned biases (garnered from the IAT) towards US Spanish features in semi-implicit attitudes (elicited from the MGT)’ - please reframe
-suggest a discussion of the potential limitations of the study in this section or the General Discussion above
References
R.M. and A. McNeill (2023) Implicit and Explicit Language Attitudes. New York/London: Routledge.
Round 2
Reviewer 1 Report
This article is much improved and I look forward to seeing the final version! See a few in-text comments.

Author Response
See attachment - thank you!
